# A Comparative Study Between the Dietary Pattern in Argentina and International Recommendations

**DOI:** 10.3390/foods14111920

**Published:** 2025-05-28

**Authors:** Carola Hunziker, Ezequiel Martín Arrieta, Maria Pilar Diaz, Alberto Edel León

**Affiliations:** 1Instituto de Investigaciones en Ciencias de la Salud (INICSA), Universidad Nacional de Córdoba (UNC), Consejo Nacional de Investigaciones Científicas y Técnicas (CONICET), Córdoba X5000, Argentina; carodavi33@gmail.com (C.H.); pdiaz@fcm.unc.edu.ar (M.P.D.); 2Escuela de Nutrición, Universidad Nacional de Córdoba, Córdoba X5000, Argentina; 3Instituto de Ciencia y Tecnología de Alimentos de Córdoba (ICYTAC), Universidad Nacional de Córdoba (UNC), Consejo Nacional de Investigaciones Científicas y Técnicas (CONICET), Córdoba X5000, Argentina; earrieta@imbiv.unc.edu.ar; 4Facultad de Ciencias Agropecuarias, Universidad Nacional de Córdoba, Córdoba X5000, Argentina

**Keywords:** environmental sustainability, food system, non-communicable diseases

## Abstract

Current food systems pose serious threats to both human health and environmental sustainability. Market forces stimulate food production to such an extent that life on Earth is endangered, and an eating pattern based on calorie-dense, nutrient-poor, and ultra-processed foods is promoted. In terms of health, this situation has caused obesity to become a pandemic, which is a major risk factor for non-communicable diseases, one of the current leading causes of death. The EAT-Lancet Commission (EAT-LC) argues that providing a growing world population with healthy diets through sustainable food systems is an immediate challenge. Thus, we propose to compare: (a) the consumption patterns described by this commission for Latin America and the Caribbean (LA&C) with those determined for Argentina; (b) Argentine dietary habits with the healthy and sustainable dietary pattern proposed by the EAT-LC; and (c) the availability of each food group in Argentina with the pattern proposed by the EAT-LC. A total of 11 food groups proposed by the Commission were taken as a reference. The dietary patterns described by the EAT-LC for LA&C are not representative of Argentina, where meat consumption is much higher (200%) and legume consumption is lower (78%). Very marked differences are observed between the consumption pattern in Argentina and the recommendations made by the EAT-LC. Regarding food availability, Argentina exceeds the requirements for meat (13.4 times), starchy vegetables (2.4 times), and eggs (2.3 times); by contrast, it does not reach the levels proposed for other products such as vegetables, nuts, legumes, or fish. Promoting a change in the eating habits of the population from a human-rights perspective represents an enormous challenge. The main aim of this work is to take a first step towards that transformation by comparing international recommendations for health and sustainability—paying special attention to culture-specific aspects—and dietary patterns in Argentina.

## 1. Introduction

Over the past few decades, dietary patterns have changed dramatically around the world. Despite regional differences, societies have generally increased their consumption of refined grains, sugar, salt, fats, and animal-source foods, while decreasing their intake of fruits, vegetables, legumes, and whole grains. Several factors have contributed to these rapid transformations, including globalization, urbanization, rising incomes, and the erosion of home cooking habits due to changes in women’s labor participation. This shift in dietary patterns has had a profound impact on global public health, as unhealthy diets are a major risk factor for non-communicable diseases (NCDs), including cardiovascular disease, cancer, type 2 diabetes, and other conditions associated with obesity [1]. In fact, one in every five deaths worldwide is attributable to poor dietary quality [2].

Beyond its critical role in human health, food consumption and production are also major drivers of environmental sustainability [3]. Due to its scale, agriculture is among the human activities with the highest environmental impact, contributing significantly to the transgression of planetary boundaries [4]. This is primarily due to the conversion of natural ecosystems into croplands and pastures, biodiversity loss, soil degradation, and water pollution from both over- and under-fertilization [5]. Population growth, demographic shifts, and increasing household food waste are expected to exacerbate these impacts, unless substantial changes occur in how food is produced, processed, and consumed [6].

Argentina, a middle-income country, is currently facing a triple burden of malnutrition: undernutrition, micronutrient deficiencies, and a high prevalence of overweight, obesity, and diet-related NCDs [7]. Alarmingly, recent estimates indicate that only 6% of the population meets the recommended intake of fruits and vegetables [8], and most people fall short in their consumption of other key healthy food groups such as legumes, fish, nuts, and seeds [9]. However, just one-third of the population acknowledges following an unhealthy diet [8]. Concurrently, Argentina ranks among the highest in Latin America for per capita sales of ultra-processed foods (UPFs) [7]. The country also has one of the highest per capita intakes of animal-source foods—especially beef—at levels that surpass those of many high-income countries [10]. Given the substantial environmental demands of livestock production [11], the ecological footprint of the Argentine diet is considerable, particularly in terms of land use and greenhouse gas (GHG) emissions [12].

Shifting diets toward healthier and more sustainable directions is one of the most pressing challenges of our time. National food-based dietary guidelines (FBDGs) offer a critical opportunity to reduce both environmental impacts and diet-related NCDs [13] However, the development of culturally acceptable guidelines requires an understanding of regional dietary differences [14]. In Argentina, there is a notable lack of comparative studies addressing this issue. Therefore, the aim of this study is to compare the diet in one Argentine province (Córdoba) with national-level data. In addition, we explore the gap between actual food consumption habits and international recommendations for healthy and sustainable diets.

## 2. Materials and Methods

We used secondary data from the Lancet studies, balance sheets [15], and dietary patterns identified and published by the Córdoba Group of Environmental Epidemiology of Cancer (GEACC), Argentina [16], a research group belonging to the Institute of Health Sciences Research, dependent on the National University of Córdoba and CONICET, and the National Survey of Household Income and Expenditure (NSHIE), which is a national survey that collects information on Argentine household incomes and expenditures [17].

For this work, the following data were used from these sources:

Healthy reference diet from EAT-Lancet Commission (EAT-LC): We followed the EAT-LC’s suggestions and recommendations for healthy eating. The EAT-LC systematizes a healthy and sustainable diet around 11 food groups, which serves as a reference. The diet includes the average daily value (in grams) for each food group, necessary to meet the nutritional and environmental requirements of the population over 2 years of age. Using several approaches, the EAT-LC estimates the effect of the reference diet on premature mortality. The 11 food groups are red meat, starchy vegetables, eggs, poultry, dairy, fish, vegetables, fruits, legumes, whole grains, and nuts.

Diet of the Central Region of Argentina (GEACC): This research paper used a structured questionnaire, which included self-reported information on socio-demographic characteristics and dietary behaviors. In particular, dietary exposure was investigated through two validated tools, a food frequency questionnaire (FFQ) on 127 foods and the support of a photographic atlas. The face-to-face interview was conducted by specially trained interviewers on 1390 subjects [18].

In this paper, we extracted 127 individual foods and classified them into 11 food group categories according to the EAT-LC recommendations; then, we calculated the per capita average daily consumption of each food group in grams. The food groups were constituted as shown in Appendix A.

National Survey of Household Income and Expenditure (NSHIE): The NSHIE includes a nationally representative sample of 21,547 households located in urban settlements with more than 2000 people. Each household recorded information on 376 food and beverage items, consumed at home and away from home, purchased over the course of 7 days in a structured self-administered questionnaire. Foods were categorized into the same groups as GEACC. Datasets are publicly available on the INDEC website (https://www.indec.gob.ar/indec/web/Nivel4-Tema-4-45-151) (accessed on 14 May 2025).

Food Balance Sheet (FBS): This shows the total quantity of food produced plus the total quantity imported and is adjusted for any changes that may have occurred in stocks to provide the available domestic supply during the reference period. The supply per person of each food product available for human consumption is calculated by dividing the corresponding quantity by the data of the population that consumes them. The food groups of the balance sheet for the Argentine population were systematized according to the groups proposed by EAT-LC; then, the daily per capita availability of each food group in grams was calculated [19]. The food groups are classified as shown in Appendix A.

### 2.1. Data Processing

First, current food consumption in Latin America and the Caribbean (LA&C) was compared with the average diet in Argentina (NSHIE) and in the province of Córdoba (GEACC). To facilitate this comparison, the average intake in LA&C was used as the reference point. Second, the recommended intake levels for a healthy and sustainable diet proposed by the EAT-LC were compared with the actual consumption in Argentina (NSHIE) and Córdoba (GEACC). Finally, the EAT-LC reference values were compared with the per capita availability of food in Argentina, based on national food balance sheets.

### 2.2. Statistical Analysis

Measures of central tendency (arithmetic mean) were estimated to extract the average daily per capita consumption in grams of the different food groups; the variables were quantitative. The statistical programme Stata v. 17 was used.

## 3. Results

The study conducted by EAT-LC establishes the current diets that are verified in the different regions of the world. Figure 1 illustrates the dietary patterns established by EAT-LC for LA&C, standardized as 100%, compared with those of Argentina, estimated by GEACC [16] and by NSHIE [17].

The distributions of average food consumption based on the EAT-LC recommendations and those obtained from the estimation of Argentina’s dietary patterns are shown in Figure 2. Indeed, red meat should be consumed only 6.1–7.6% and poultry only 17.8–21.1% of the current amount. By contrast, certain foods are consumed in a much lower percentage than the recommended amounts; the Argentine population consumes only 5.3–6.7% of the recommended legumes and 25–60% of the recommended fish. While the values for the consumption of cereal-based products are similar to those suggested by EAT-LC, around 95 % of the wheat milled in Argentina is used for the production of refined flours [19], another problem of the Argentine dietary pattern. In summary, the data revealed an overconsumption of red meat and poultry, and an intake below recommended levels for fish.

A key aspect of studying the possibilities of a country to modify its eating habits is the analysis of food availability. Figure 3 compares EAT-LC dietary recommendations with availability in Argentina according to the balance sheet of the different food groups.

## 4. Discussion

In the present study, the dietary patterns assigned by the EAT-LC to the LAC were compared with what is consumed in Argentina by two methods, the dietary patterns identified GEACC [16] and the NSHIE [17]. Significant differences were observed between EAT-LC’s suggestions and Argentina’s consumption patterns, which served as a starting point to analyze whether the foods suggested by the EAT-LC were available in our country.

Although the two ways of estimating consumption in Argentina show the same trend, when analyzing peoples’ answers on what they consumed (GEACC), we obtained a healthier profile than when we analyzed their answers on the foods they purchased (NSHIE). In the first case, higher values were observed for fish, legumes, fruits, and vegetables, and lower values for red meat and starchy vegetables. These results are in line with the Risk Factors Survey, revealing that 61.6% of the population is overweight, but 72% believe that their diet is healthy [21].

In any case, the consumption pattern for LA&C reported by EAT-LC exhibited great differences with the consumption pattern in Argentina, where the consumption of red meat is more than 200% higher, poultry 100% higher, and vegetables almost 150% higher than in the rest of Latin America and the Caribbean region. Argentina’s dietary patterns show significant differences compared to other Latin American and Caribbean countries, particularly in beef consumption. The country has a deeply rooted cattle farming tradition, which has led to a high per capita meat intake compared to the rest of the region. Factors such as the availability of suitable land for livestock farming, the cultural significance of “asado” (barbecue), and the historical preference for red meat have reinforced this dietary pattern [22]. In contrast, the consumption of starchy vegetables is lower and that of legumes is about 75% lower in Argentina than in the rest of the region. In the Argentine control population, the consumption of eggs is approximately 25% less than recommended, while the intake of dairy products is 50% less than in the rest of the region. Grain consumption in Argentina exceeds the values reported for LA&C, as reported by both GEACC and NSHT. However, 81% of the GEACC estimation and 95% of the NSHT assessment corresponded to products from refined grains, not from whole wheat flours as recommended by EAT-LC. As regards the remaining food groups covered in this analysis, no significant differences or variations were evident between the two estimation methods. The diet described for LA&C is not representative of Argentina. In this sense, Matos and Brandani [23] emphasize that the dietary habits of the Argentine population present certain particularities that differentiate it from those in other Latin American countries, especially due to the high consumption of red meat. Meat consumption in Argentina is deeply embedded in the country’s cultural identity and economic structure, posing a significant barrier to its reduction. The nation’s strong cattle farming tradition, the social practice of asado (barbecue), and the perception of meat as an essential food have reinforced consumption habits that are difficult to alter. Moreover, Argentina’s agricultural economy heavily relies on beef production, which leads to resistance against policies aimed at transitioning toward more sustainable diets [24]. According to Onwezen and Dagevos [25], “barriers to reducing meat consumption include motivational, emotional, and structural factors, such as the availability and visibility of plant-based options in food environments”. Overcoming these challenges requires the implementation of viable policy measures, such as incentives for the production and commercialization of alternative proteins, awareness campaigns highlighting the environmental and health benefits of a reduced-meat diet, and menu restructuring in public spaces to improve the accessibility of plant-based options. This explains why we did not frame the consumption pattern of Argentina within the parameters of LA&C and, instead, studied its relationship with EAT-LC’s parameters.

Matos and Brandani [23] confirm this trend in Argentine dietary consumption, pointing out that the traditional dietary profile is characterized by a high consumption of animal proteins and fats obtained mainly from red meat, and a low intake of fish, fruit, and vegetables. A previous study [7] showed coincidences in terms of the dietary patterns detected: the traditional dietary pattern representative of the population of the province of Córdoba was characterized by high factor loads of fatty meats, eggs, and refined cereals. Moreover, the latest National Nutrition and Health Survey in Argentina further corroborated these results and concluded that the proportion of the population reporting daily or frequent consumption of non-recommended foods (snack products, refined flours, and sugary drinks, among others) was alarmingly high considering their high sugar, fat and salt contents and low nutritional value. Concerning the high consumption of refined flours and low consumption of whole grains and legumes, results agree with the work of the Centre for Studies on Child Nutrition (CESNI), which examined changes in the pattern of food consumption between 1996 and 2013 and concluded that legumes consumption was halved, pastries and biscuits consumption increased, and empanadas, sandwiches, cakes, and pizza quadrupled [26]. The essential transition towards a healthier and more sustainable diet should be understood as a as a guide rather than a standardization leading the almost 8 billion people on the planet to consume the same food and, thus, to lose their cultural identity [12].

The availability of certain products exceeds EAT-LC requirements: red meat is 13.4 times more available, starchy vegetables are 2.4 times more available, eggs are 2.3 times more available, poultry 3.6 times more available, dairy products 2.1 times more available, and whole grains 2.5 times more available. By contrast, some products do not meet the availability requirements or are not available at all: only 63% of the vegetables and 67.8% of fish recommended by the EAT-LC are available.

In this sense, the low consumption versus the vast production of whole grains in Argentina is striking. Besides, Argentina presents an excess consumption of refined bakery products from refined flours. The survey conducted by CESNI on this trend highlights the urgent need to reorient food education actions to promote the consumption of whole grains [26].

## 5. Conclusions

Argentina exhibits a diet that significantly differs from the guidelines established by EAT-LC for Latin America and the Caribbean, characterized by excessive red meat consumption and a low intake of legumes, fish, and nuts. While EAT-LC recommendations provide a general framework for transforming Argentina’s dietary model, their implementation requires specific strategies. In this regard, fostering cultural change through educational campaigns is essential to reduce the consumption of meat and refined flours while increasing the intake of whole grains, legumes, fish, and nuts. Additionally, overcoming access barriers to these foods through public policies aimed at ensuring their affordable availability for the entire population is crucial.

In this context, Argentina’s 2021 Healthy Eating Promotion Law (Law 27.642) serves as a key instrument to address this issue. Its intersectoral approach reinforces consumers’ rights to information by requiring labeling of critical nutrients in packaged products and promoting food and nutritional education as a structural change strategy.

This study presents certain limitations, including the use of secondary data sources, which may affect the precision of individual consumption estimates. Nonetheless, the findings contribute to the discussion on the sustainability and adequacy of Argentina’s dietary patterns, emphasizing the need for multidisciplinary research to define parameters for a healthy, equitable, and accessible diet within a rights-based approach.

## Figures and Tables

**Figure 1 foods-14-01920-f001:**
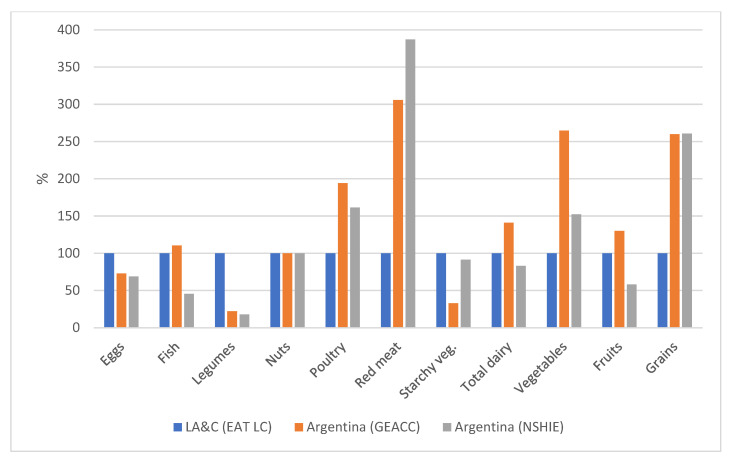
Relationship among the consumption of different food groups in Argentina, estimated by GEACC [16] and by NSHIE [17] and the consumption in Latin American and Caribbean region (LA&C) according to EAT-LC [15], standardized as 100%.

**Figure 2 foods-14-01920-f002:**
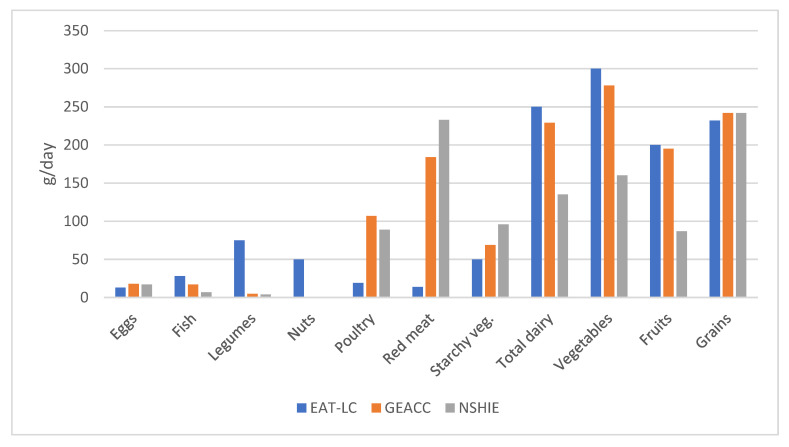
Comparison among the consumption pattern suggested by the Lancet Commission (EAT-LC) [15] and the consumption of different food groups in Argentina, estimated by GEACC [16] and by NSHIE [17].

**Figure 3 foods-14-01920-f003:**
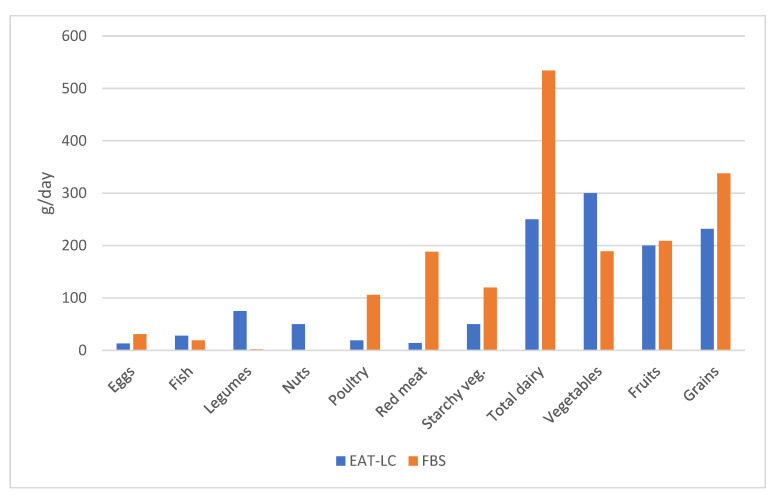
Comparison between dietary recommendations proposed by the Lancet Commission (EAT-LC) [15] and food availability in Argentina according to the food balance sheet (FBS) [20].

## Data Availability

The original contributions presented in the study are included in the article/Appendix A. Further inquiries can be directed to the corresponding author.

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
