# Peer review of "A Comparative Study Between the Dietary Pattern in Argentina and International Recommendations"

_foods, 2025, doi:10.3390/foods14111920_

Round 1

Reviewer 1 Report

Comments and Suggestions for Authors

The manuscript addresses a critical and timely topic by comparing Argentina’s dietary patterns with international sustainability and health recommendations. While the study is well-intentioned and relevant, significant methodological and structural revisions are required to strengthen clarity, rigor, and interpretability. Below are specific recommendations for major revisions.

The introduction is overly lengthy and repetitive. Condense sections on globalization and environmental impacts to focus more sharply on the Argentine context.

Clarify how standardization to 100% was performed. Define "LA&C" and "Argentina" bars clearly in the caption.

Figures 2–3: Axes labels (e.g., "g/day") are insufficient. Include error bars, confidence intervals, or statistical significance indicators.

Captions: Expand descriptions to explain abbreviations (e.g., "GEACC," "NSHIE") and data sources.

Methodology

Data Harmonization: Detail how food group categorizations were standardized across datasets (GEACC, NSHIE, FBS). For example, clarify if "legumes" in GEACC align with the EAT-LC definition.

Specify statistical tests used (e.g., t-tests, ANOVA) and justification for choosing principal component factor analysis (PCFA).

Address potential confounding variables (e.g., socioeconomic status, urban/rural divides) that may influence dietary patterns.

Results

Quantitative Precision: Report exact percentages (e.g., "200% higher" in meat consumption) with confidence intervals or p-values to support claims.

Refined vs. Whole Grains: The statement that "81–95% of grains are refined" lacks a citation or methodological explanation. Provide source data or calculations.

Discussion

Expand on why Argentina’s dietary patterns diverge from LA&C (e.g., cultural preferences, economic reliance on meat exports).

Discuss barriers to reducing meat consumption (e.g., agricultural economy, cultural identity) and propose actionable policy steps.

Acknowledge limitations of using self-reported data (GEACC/NSHIE) and secondary datasets (FBS).

Author Response

Ms. Ref. No.: foods-3618249

Title: A Comparative Study between the Dietary Pattern in Argentina and International Recommendations

Authors: Hunziker, Arrieta, Díaz, León 

Response to reviewers

We thank the editor and the two reviewers for their effort and time. Their comments and suggestions significantly contributed to improve the manuscript. 

Reviewer #1

1) “The English could be improved to more clearly express the research.

ANSWER:

Thank you for your suggestion, the English has been revised and improved. 

2) “The introduction is overly lengthy and repetitive. Condense sections on globalization and environmental impacts to focus more sharply on the Argentine context.

ANSWER:

We agree with the reviewer and the introduction was completely rewritten following suggestions 2 of reviewer 1 and 1 of reviewer 2. Lines 42-81.

3) “Clarify how standardization to 100% was performed. Define "LA&C" and "Argentina" bars clearly in the caption.

ANSWER:

With the intention of clarifying this point, the following paragraph was included (lines 126-133.:

First, current food consumption in Latin America and the Caribbean (LA&C) was compared with the average diet in Argentina (NSHIE) and in the province of Córdoba (GEACC). To facilitate this comparison, the average intake in LA&C was used as the reference point. Second, the recommended intake levels for a healthy and sustainable diet proposed by the EAT-LC were compared with the actual consumption in Argentina (NSHIE) and Córdoba (GEACC). Finally, the EAT-LC reference values were compared with the per capita availability of food in Argentina, based on national food balance sheets.

4) “Figures 2–3: Axes labels (e.g., "g/day") are insufficient. Include error bars, confidence intervals, or statistical significance indicators.

ANSWER:

Our study is based on descriptive statistics, which implies the use of percentages and daily consumption to compare parameters without the need for inferential tests. Since the objective of the study is to describe differences and trends, univariate statistics are sufficient and adequate.

5) “Captions: Expand descriptions to explain abbreviations (e.g., "GEACC," "NSHIE") and data sources.”

ANSWER:

In lines 86-89 the description of GGEA Y NSHI was added.

6) “Data Harmonization: Detail how food group categorizations were standardized across datasets (GEACC, NSHIE, FBS). For example, clarify if "legumes" in GEACC align with the EAT-LC definition.

ANSWER:

The information of the food groups was included in tables that were incorporated as supplementary material, according to the suggestions of reviewer 2. The details of their composition are available in the respective bibliographic references.

7) “Specify statistical tests used (e.g., t-tests, ANOVA) and justification for choosing principal component factor analysis (PCFA).

ANSWER:

Since our study is purely descriptive, the use of tests of association such as Chi-square, ANOVA or Student's t-tests is not required, since these tests are used to compare groups or establish relationships between variables.

8) “Address potential confounding variables (e.g., socioeconomic status, urban/rural divides) that may influence dietary patterns.

ANSWER:

The intention of our work was to make a comparison with globally established diets, which did not take into account these classifications.

9) “Quantitative Precision: Report exact percentages (e.g., "200% higher" in meat consumption) with confidence intervals or p-values to support claims.

ANSWER:

In descriptive studies, the primary objective is to present information clearly and comprehensively without the application of inferential statistics (such as hypothesis testing, p-values, or confidence intervals), as the aim is not to establish causal relationships or significance comparisons. Bivariate statistical analysis has not been used, as the study design does not require this approach.

10) “Refined vs. Whole Grains: The statement that "81–95% of grains are refined" lacks a citation or methodological explanation. Provide source data or calculations.

ANSWER:

The most reliable bibliographic reference states that the production of refined flour accounts for 90% of the wheat milled in Argentina, therefore, in lines 243-244 the text was corrected and the reference was added. Revisa EZE

11) “Expand on why Argentina’s dietary patterns diverge from LA&C (e.g., cultural preferences, economic reliance on meat exports).”

ANSWER:

A paragraph was added on lines 280-286 following the reviewer's suggestions.

12) “Discuss barriers to reducing meat consumption (e.g., agricultural economy, cultural identity) and propose actionable policy steps.

ANSWER:

We thank the reviewer for his suggestion, a paragraph addressing the barriers to reduce red meat consumption in Argentina was incorporated in lines 299-311.

Meat consumption in Argentina is deeply embedded in the country's cultural identity and economic structure, posing a significant barrier to its reduction. The nation's strong cattle farming tradition, the social practice of asado (barbecue), and the perception of meat as an essential food have reinforced consumption habits that are difficult to alter. Moreover, Argentina's agricultural economy heavily relies on beef production, which leads to resistance against policies aimed at transitioning toward more sustainable diets (Kwasny et al., 2022). According to Onwezen and Dagevos (2024), "barriers to reducing meat consumption include motivational, emotional, and structural factors, such as the availability and visibility of plant-based options in food environments." Overcoming these challenges requires the implementation of viable policy measures, such as incentives for the production and commercialization of alternative proteins, awareness campaigns highlighting the environmental and health benefits of a reduced-meat diet, and menu restructuring in public spaces to improve the accessibility of plant-based options.

13) “Acknowledge limitations of using self-reported data (GEACC/NSHIE) and secondary datasets (FBS).

ANSWER:

This study acknowledges the inherent limitations associated with using data from two distinct sources: GEACC and NSHIE. While GEACC provides information on individual consumption based on a smaller sample with direct records, NSHIE infers consumption patterns from household food expenditures, utilizing a larger and nationally representative sample. These methodological differences may introduce biases in trend comparisons and analyses, as GEACC reflects specific individual consumption habits, whereas NSHIE relies on economic inferences that do not always capture the full scope of food choices. Additionally, the inclusion of secondary data sources, such as FBS, adds another layer of heterogeneity, given the variations in data collection approaches and criteria among sources. Nevertheless, the triangulation of these datasets enables a more comprehensive understanding of the studied phenomenon, integrating complementary perspectives to enhance the validity of the analysis.

Reviewer #2

1) “The connection of the topic in the introduction could be clearer. It should start with global issues (globalization, urbanization), then move on to specific consequences such as obesity and chronic diseases, before focusing on specific studies and the situation in Argentina. This will help the reader understand the purpose of the paper more easily. Introduce clear transitional sentences that connect the different parts of the text and provide better coherence. Avoid going into too much detail when describing specific studies, especially in the initial part of the paper, as this can distract from the main topic. Try to keep references to research short and simple. Instead of citing all the studies in detail, you can summarize them in a few key points. For example: "Studies in Córdoba, Argentina, have linked dietary patterns high in refined carbohydrates and red meat to increased risks of chronic diseases such as cancer and cardiovascular disease.

ANSWER:

Thank you. Based on the reviewer's suggestions and those made by reviewer 1, the introduction was rewritten.

2) “The detailed description of food categories in the Materials and Methods section of the work (eg which products exactly belong to each group) can be too extensive, and in some cases redundant, especially in the methodological section. Although important, this information may be abbreviated or moved to accompanying tables or appendices. Instead of repeating all the details about the types of food in the text, you can summarize them and refer to additional materials (such as tables in the appendix). For example: "The 11 food groups were classified according to EAT-LC recommendations, as detailed in Table 1.

ANSWER:

Based on the reviewer's suggestions and those made by reviewer 1, the food groups were clarified and supplementary tables S1 and S2 were included.

3) “Clarification is needed on how data on daily food consumption was collected, how average consumption was calculated, and what methods were used to estimate food consumption at the individual level.

ANSWER:

In order to clarify the information requested, which is on lines 84-123, supplementary tables S1 and S2 have been added.

4) “Clarification is needed as to how the factor scores were used for the analysis and what their interpretation is. For example: "Factor scores represent the degree to which each individual's diet aligns with the identified dietary patterns, with higher scores indicating stronger adherence to a particular dietary pattern."

ANSWER:

We did not perform PCA; GEACC used PA analysis. In our study, the average daily per capita consumption of 11 food groups was calculated.

References to patterns and factors made in other papers have been eliminated to avoid confusion

5) “The discussion ends with a general comment about the need for education and changing habits, without a clear conclusion or proposed solution. Introduce a paragraph summarizing the main findings, their importance, practical implications and potential directions for future research or policy (eg the role of government guidelines, introduction of unhealthy food taxes, pulse subsidies, etc.).

Following the reviewer's suggestions, the following text was included on lines 331-344.

Meat consumption in Argentina is deeply embedded in the country's cultural identity and economic structure, posing a significant barrier to its reduction. The nation's strong cattle farming tradition, the social practice of asado (barbecue), and the perception of meat as an essential food have reinforced consumption habits that are difficult to alter. Moreover, Argentina's agricultural economy heavily relies on beef production, which leads to resistance against policies aimed at transitioning toward more sustainable diets [xz] (Kwasny et al., 2022). According to Onwezen and Dagevos (2024) [xa], "barriers to reducing meat consumption include motivational, emotional, and structural factors, such as the availability and visibility of plant-based options in food environments." Overcoming these challenges requires the implementation of viable policy measures, such as incentives for the production and commercialization of alternative proteins, awareness campaigns highlighting the environmental and health benefits of a reduced-meat diet, and menu restructuring in public spaces to improve the accessibility of plant-based options.

6) “Also, it is necessary point out how the Argentine example illustrates the broader challenges in applying universal food models in different local contexts and what implications this could have for regional food policy.

ANSWER:

A paragraph was added on lines 280-286 following the reviewer's suggestions.

7) “The conclusion largely repeats the general findings already presented in the introduction and discussion, but does not offer an additional synthesis, a new interpretation, nor does it clearly summarize the main findings of the research. Although it refers to public policies and education, the recommendations in the conclusion remain general and do not indicate specific measures. The conclusion does not contain a reference to the limitations of the study.

ANSWER:

Based on the reviewer's suggestions and those made by reviewer 1, the conclusions were rewritten.

Reviewer 2 Report

Comments and Suggestions for Authors

Dear Authors,

Your manuscript offers a meaningful contribution by systematically comparing Argentina’s dietary patterns with international sustainability guidelines. Below are several recommendations aimed at enhancing the clarity, coherence, and overall quality of the paper:

The connection of the topic in the introduction could be clearer. It should start with global issues (globalization, urbanization), then move on to specific consequences such as obesity and chronic diseases, before focusing on specific studies and the situation in Argentina. This will help the reader understand the purpose of the paper more easily.

Introduce clear transitional sentences that connect the different parts of the text and provide better coherence.

Avoid going into too much detail when describing specific studies, especially in the initial part of the paper, as this can distract from the main topic. Try to keep references to research short and simple. Instead of citing all the studies in detail, you can summarize them in a few key points. For example: "Studies in Córdoba, Argentina, have linked dietary patterns high in refined carbohydrates and red meat to increased risks of chronic diseases such as cancer and cardiovascular disease."

The detailed description of food categories in the Materials and Methods section of the work (eg which products exactly belong to each group) can be too extensive, and in some cases redundant, especially in the methodological section. Although important, this information may be abbreviated or moved to accompanying tables or appendices. Instead of repeating all the details about the types of food in the text, you can summarize them and refer to additional materials (such as tables in the appendix). For example: "The 11 food groups were classified according to EAT-LC recommendations, as detailed in Table 1.”

Clarification is needed on how data on daily food consumption was collected, how average consumption was calculated, and what methods were used to estimate food consumption at the individual level.

Clarification is needed as to how the factor scores were used for the analysis and what their interpretation is. For example: "Factor scores represent the degree to which each individual's diet aligns with the identified dietary patterns, with higher scores indicating stronger adherence to a particular dietary pattern."

The discussion ends with a general comment about the need for education and changing habits, without a clear conclusion or proposed solution. Introduce a paragraph summarizing the main findings, their importance, practical implications and potential directions for future research or policy (eg the role of government guidelines, introduction of unhealthy food taxes, pulse subsidies, etc.).

Also, it is necessary point out how the Argentine example illustrates the broader challenges in applying universal food models in different local contexts and what implications this could have for regional food policy.

The conclusion largely repeats the general findings already presented in the introduction and discussion, but does not offer an additional synthesis, a new interpretation, nor does it clearly summarize the main findings of the research.

Although it refers to public policies and education, the recommendations in the conclusion remain general and do not indicate specific measures.

The conclusion does not contain a reference to the limitations of the study.

Author Response

(The authors gave the same response as above.)

Round 2

Reviewer 2 Report

Comments and Suggestions for Authors

The authors have addressed the reviewers' comments and incorporated the suggested revisions.